# All-trans retinoic acid induces synaptopodin-dependent metaplasticity in mouse dentate granule cells

**Maximilian Lenz[1], Amelie Eichler[1], Pia Kruse[1], Julia Muellerleile[2], Thomas Deller[2], Peter Jedlicka[2,3][†], Andreas Vlachos[1,4,5]\*[†]**

[1]Department of Neuroanatomy, Institute of Anatomy and Cell Biology, Faculty of Medicine, University of Freiburg, Freiburg, Germany; [2]Institute of Clinical Neuroanatomy, Dr. Senckenberg Anatomy, Neuroscience Center, Goethe-University Frankfurt, Frankfurt am Main, Germany; [3]ICAR3R - Interdisciplinary Centre for 3Rs in Animal Research, Faculty of Medicine, Justus-Liebig-University, Giessen, Germany; [4]Center for Basics in Neuromodulation (NeuroModulBasics), Faculty of Medicine, University of Freiburg, Freiburg, Germany; [5]Center Brain Links Brain Tools, University of Freiburg, Freiburg, Germany

**\*For correspondence:**
andreas.vlachos@anat.uni-freiburg.de

[†]These authors are joint senior authors to this work

**Abstract** Previously we showed that the vitamin A metabolite all-trans retinoic acid (atRA) induces synaptic plasticity in acute brain slices prepared from the mouse and human neocortex (Lenz et al., 2021). Depending on the brain region studied, distinct effects of atRA on excitatory and inhibitory neurotransmission have been reported. Here, we used intraperitoneal injections of atRA (10 mg/kg) in adult C57BL/6J mice to study the effects of atRA on excitatory and inhibitory neurotransmission in the mouse fascia dentata—a brain region implicated in memory acquisition. No major changes in synaptic transmission were observed in the ventral hippocampus while a significant increase in both spontaneous excitatory postsynaptic current frequencies and synapse numbers were evident in the dorsal hippocampus 6 hr after atRA administration. The intrinsic properties of hippocampal dentate granule cells were not significantly different and hippocampal transcriptome analysis revealed no essential neuronal changes upon atRA treatment. In light of these findings, we tested for the metaplastic effects of atRA, that is, for its ability to modulate synaptic plasticity expression in the absence of major changes in baseline synaptic strength. Indeed, in vivo long-term potentiation (LTP) experiments demonstrated that systemic atRA treatment improves the ability of dentate granule cells to express LTP. The plasticity-promoting effects of atRA were not observed in synaptopodin-deficient mice, therefore, extending our previous results regarding the relevance of synaptopodin in atRA-mediated synaptic strengthening in the mouse prefrontal cortex. Taken together, our data show that atRA mediates synaptopodin-dependent metaplasticity in mouse dentate granule cells.

## Introduction

Adaptive processes in synaptic sites of the central nervous system are fundamental to normal brain function (*Citri and Malenka, 2008*; *Humeau and Choquet, 2019*; *Neves et al., 2008*). Several major signaling pathways and mechanisms that mediate and modulate synaptic plasticity have been identified in the past decades (*Abraham and Bear, 1996*; *Lüscher and Malenka, 2012*; *Turrigiano, 2008*). One of the key mediators of excitatory and inhibitory synaptic plasticity among plasticity-related signaling molecules is the vitamin A derivative all-trans retinoic acid (atRA) (*Chen et al., 2014*; *Aoto et al., 2008*; *Sarti et al., 2013*; *Yee and Chen, 2016*; *Zhang et al., 2018*; *Shibata et al., 2021*).

Prior to this study, we showed that atRA potentiates excitatory postsynapses in human cortical slices prepared from neurosurgical access tissue (*Lenz et al., 2021*). This observation is consistent with previous reports suggesting that atRA mediates the accumulation of AMPA receptors at synaptic sites (*Poon and Chen, 2008*; *Maghsoodi et al., 2008*). Moreover, we demonstrated that the presence of the plasticity-related protein synaptopodin, which is an essential component of the calcium ion-storing spine apparatus organelle (*Deller et al., 2003*; *Mundel et al., 1997*; *Vlachos et al., 2013*; *Vlachos et al., 2009*), is required for atRA-mediated strengthening of excitatory neurotransmission in the mouse medial prefrontal cortex. In accordance with these findings, atRA triggers structural changes of synaptopodin clusters, spine apparatuses, and dendritic spines in human cortical slices (*Lenz et al., 2021*).

Thus, both atRA and synaptopodin have been firmly linked to the ability of neurons to express distinct forms of synaptic plasticity (*Chen et al., 2014*; *Jedlicka and Deller, 2017*; *Vlachos, 2012*) and their relevance in homeostatic and associative synaptic plasticity is well established. Recent reports have implicated atRA and synaptopodin in metaplasticity (*Hsu et al., 2019*; *Maggio and Vlachos, 2018*), which refers to the ability of neurons to modify their ability to express synaptic plasticity (*Abraham and Bear, 1996*; *Abraham, 2008*). Considering the role of the hippocampal formation, more specifically the role of the dentate gyrus, in memory acquisition (*Hainmueller and Bartos, 2018*), and based on previous work regarding the role of synaptopodin in synaptic plasticity (e.g., *Galanis et al., 2021*; *Jedlicka et al., 2009*; *Paul et al., 2020*; *Vlachos et al., 2008*; *Yap et al., 2020*), we studied the effects of atRA on synaptic transmission and plasticity and its link to synaptopodin in mouse dentate granule cells.

## Results

### All-trans retinoic acid has no major effects on synaptic strength and intrinsic cellular properties of dentate granule cells in the dorsal hippocampus

Adult male C57BL/6J mice were injected intraperitoneally with atRA (10 mg/kg) or vehicle-only solution, and acute coronal slices containing the dorsal hippocampus were prepared 6 hr later. AMPA-receptor-mediated spontaneous excitatory postsynaptic currents (sEPSCs) were recorded from mature granule cells in the suprapyramidal blade of the dentate gyrus (*Figure 1A–C*). In contrast to neocortical neurons (*Lenz et al., 2021*), atRA had no apparent effects on the mean sEPSC amplitude, whether half width or area (*Figure 1D*). However, a significant increase in sEPSC frequencies was observed in the atRA group (*Figure 1E*).

Subsequently, we recorded GABA-receptor-mediated spontaneous inhibitory postsynaptic currents (sIPSCs) from dentate granule cells, and we found no significant differences between the two groups (*Figure 1C, F and G*). The mean sIPSC amplitude—whether half width or area—and the sIPSC frequencies were indistinguishable between the two groups. Thus, major changes in inhibitory neurotransmission in the dentate gyrus of atRA-treated mice do not explain the increased mean sEPSC frequency (c.f., *Figure 1E*).

Finally, basic intrinsic properties were assessed (*Figure 2*). Dentate granule cells from atRA-treated animals were comparable to vehicle-only injected animals. The mean resting membrane potential (*Figure 2A*) and input resistance (*Figure 2B*) were similar in both groups. In addition, action potential (AP) frequencies were not significantly altered in the atRA group in these experiments (*Figure 2C*). Taken together, these results demonstrate that atRA treatment has no major effects on synaptic strength or the basic intrinsic properties of dentate granule cells. Specifically, atRA does not affect the sEPSC amplitudes of dentate granule cells in the dorsal hippocampus (*Lenz et al., 2021*).

### All-trans retinoic acid has no significant effects on synaptic transmission and intrinsic cellular properties of dentate granule cells in the ventral hippocampus

The ability of neurons to express synaptic plasticity varies along the septotemporal axis of the hippocampus (*Vlachos et al., 2008*; *Chawla et al., 2018*; *Maggio and Segal, 2009*; *Schreurs et al., 2017*). We therefore tested for the effects of atRA on dentate granule cells in the ventral hippocampus.

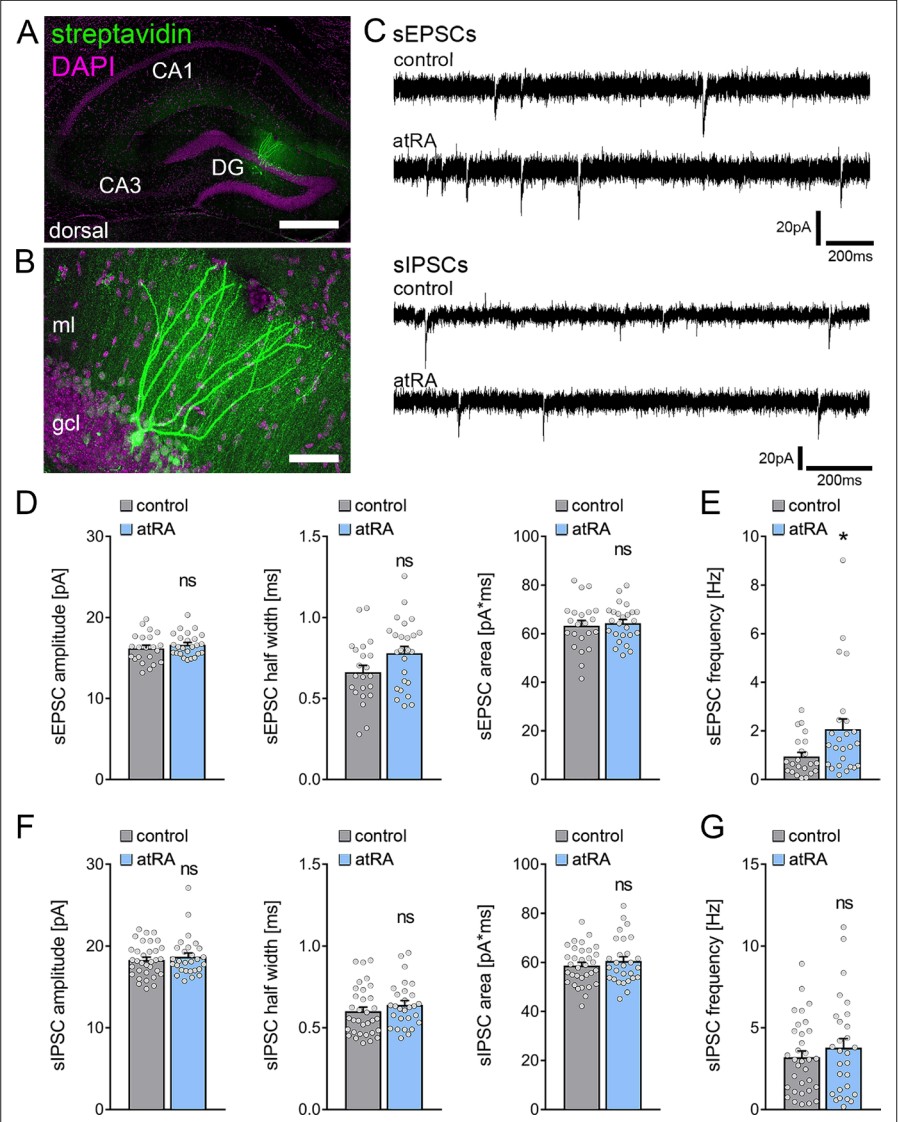

**Figure 1.** All-trans retinoic acid (atRA) induces no major changes in excitatory and inhibitory synaptic strength in dentate granule cells of the dorsal hippocampus. (**A, B**) Example of patched and post hoc identified dentate granule cell in acute slices prepared from the dorsal hippocampus. Scale bar (upper panel)=500 μm; Scale bar (lower panel)=50 μm. (**C**) Sample traces of spontaneous excitatory postsynaptic currents (sEPSCs) and spontaneous inhibitory postsynaptic currents (sIPSCs) recorded from dentate granule cells of atRA (10 mg/kg; i.p.)-treated or vehicle-only (control) animals. (**D, E**) Group data of sEPSC recordings. A significant increase in the sEPSC frequency is observed ($n_{control}$=22 cells, $n_{atRA}$=25 cells in four animals; Mann-Whitney test, $U_{sEPSC\ frequency}$=175). (**F, G**) Group data of sIPSC recordings ($n_{control}$=33 cells, $n_{atRA}$=28 cells in four animals; Mann-Whitney test). Individual data points are indicated by gray dots. Values represent mean ± SEM. (*, $p<0.05$; ns, non-significant difference). DG, dentate gyrus; gcl, granule cell layer; ml, molecular layer.

A different set of animals was injected intraperitoneally with atRA (10 mg/kg) or vehicle-only, and horizontal slices containing the ventral hippocampus were prepared 6 hr after the injection (**Figure 3A**). Neither sEPSC nor sIPSC recordings showed any significant differences between the two groups (**Figure 3B–D**). Likewise, no differences in the active and passive membrane properties were observed (**Figure 4**). Thus, we concluded that no changes in synaptic transmission and basic intrinsic properties of dentate granule cells occurred in the ventral hippocampus 6 hr after intraperitoneal atRA injection.

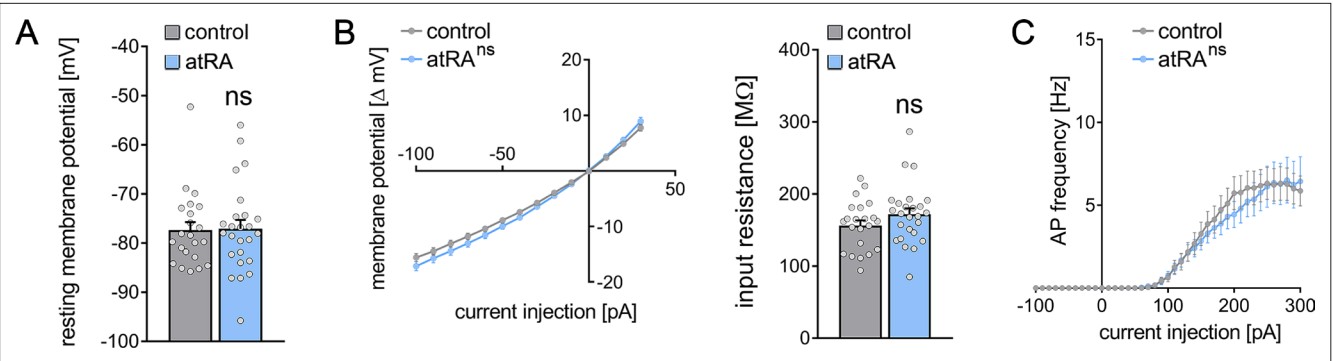

**Figure 2.** Passive or active membrane properties of dentate granule cells remain unchanged in the dorsal hippocampus following intraperitoneal administration of all-trans retinoic acid (atRA). (**A, B**) Group data of resting membrane potentials, input-output curves, and input resistances. (**C**) A slight but not significant decrease in action potential (AP) frequency of dentate granule cells is observed in the atRA group ($n_{control}$ = 22 cells, $n_{atRA}$ = 25 cells in four animals each; Mann-Whitney test for column statistics, RM two-way ANOVA followed by Sidak's multiple comparisons test for input-output curve and AP frequency analysis). Individual data points are indicated by gray dots. Values represent mean ± SEM. (ns, non-significant difference).

## All-trans retinoic acid treatment causes only limited changes in the expression of synapse-related genes in the hippocampus

Biological effects of atRA have been reported at the gene transcription level (*Maden, 2002*). To further evaluate the effects of atRA in our experimental setting transcriptome analysis was performed in hippocampal tissue samples 6 hr after intraperitoneal atRA or vehicle-only injections (*Figure 5*). Principal component analysis revealed no major clustering of samples related to the respective treatment (*Figure 5A*). In line with this observation, only a limited number of significantly regulated genes were identified (*Figure 5B*), representing a z-score heatmap clustering over treatment (29 genes; *Figure 5C*). Further analysis of the significantly regulated genes indicated that subsets of these genes relate to atRA-signaling/atRA-metabolism (~21%, 6 genes), synaptic transmission (~14%, 4 genes), or Wnt signaling (~7%, 2 genes), respectively. Notably, the majority of significantly regulated genes did not show any functional clustering (*Figure 5D*). While these findings indicate that intraperitoneally injected atRA reached and affected the hippocampus, no major changes in synaptic genes were detected 6 hr after administration of atRA.

## Increased synapse numbers in the dorsal hippocampus are detected in mice treated with all-trans retinoic acid

Next, transmission electron microscopy was used to assess the structural properties of excitatory synapses in the outer two-thirds of the molecular layer in the dorsal hippocampus which is the layer of the major excitatory input from the entorhinal cortex (*Figure 6*). Cross sections of asymmetric synapses, that is, the numbers and length of postsynaptic densities (PSDs) and presynaptic vesicle counts, were quantified in control and atRA-treated mice (*Figure 6A*). It is well established that PSD length in synaptic cross sections correlates to synaptic strength (*Meyer et al., 2014*). In agreement with our electrophysiological recordings, which showed no significant difference in the sEPSC amplitudes between the groups (c.f., *Figure 1D*), PSD lengths did not significantly change in the atRA-treated group (*Figure 6B*). However, a robust increase in the number of PSDs per area was detected, while presynaptic vesicle counts were not significantly different between the two groups (*Figure 6B and C*). These results indicate that the structural properties of synapses are not affected by atRA, and that increased synapse numbers may explain the increased sEPSC frequencies in the dorsal hippocampus of atRA-treated mice.

## All-trans retinoic acid mediates synaptopodin-dependent metaplasticity in the dentate gyrus

In light of the plasticity-promoting effects of atRA (*Aoto et al., 2008*; *Hsu et al., 2019*; *Arendt et al., 2015*), including our recent findings in the mouse and human neocortex (*Lenz et al., 2021*), we theorized that atRA could induce metaplasticity in the dentate gyrus. Specifically, the lack of essential

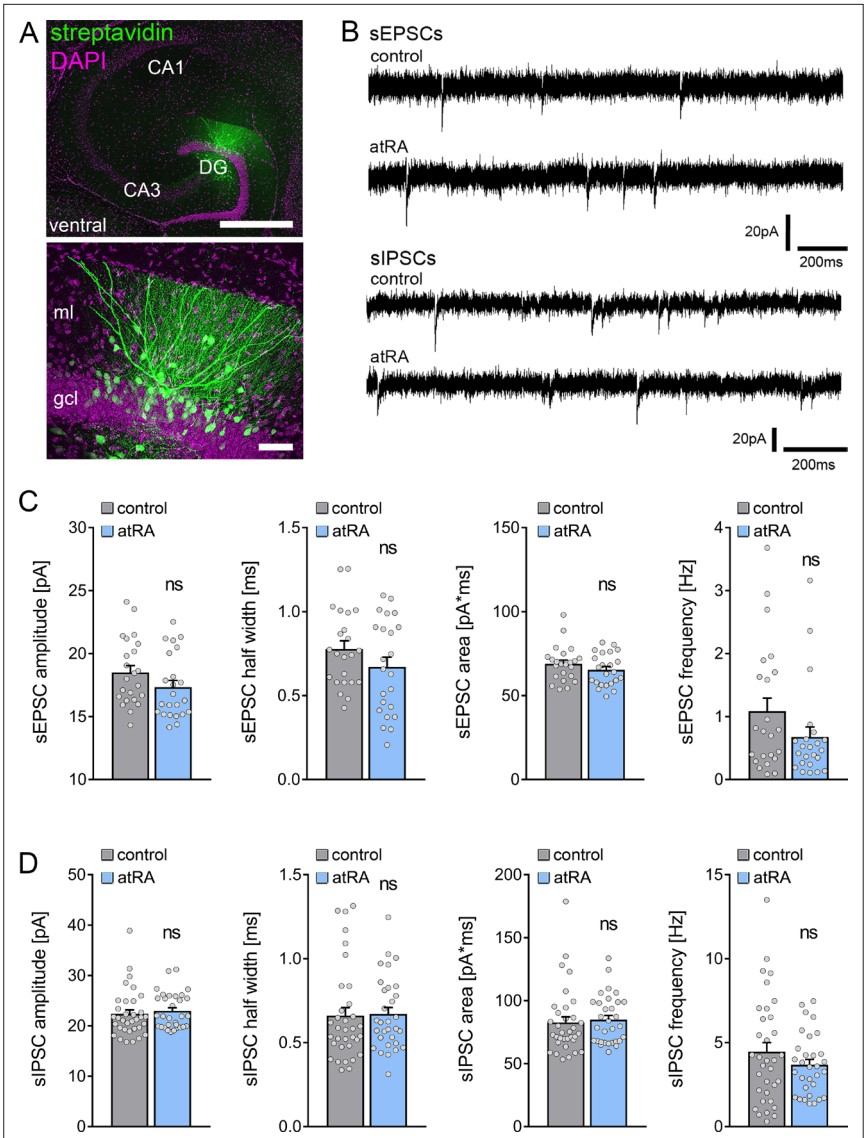

**Figure 3.** All-trans retinoic acid (atRA) does not induce changes in excitatory and inhibitory neurotransmission in the ventral hippocampus of adult mice. (**A**) Example of patched and post hoc identified dentate granule cell in acute slices prepared from the ventral hippocampus. Scale bar (upper panel)=500 μm; Scale bar (lower panel)=50 μm. (**B**) Sample traces of spontaneous excitatory postsynaptic currents (sEPSCs) and spontaneous inhibitory postsynaptic currents (sIPSCs) recorded from dentate granule cells of atRA-treated or vehicle-only (control) animals. (**C**) Group data of sEPSC recordings ($n_{control}$=23 cells, $n_{atRA}$=23 cells in four animals each; Mann-Whitney test). (**D**) Group data of sIPSC recordings ($n_{control}$=34 cells, $n_{atRA}$=31 cells in four animals each; Mann-Whitney test). Individual data points are indicated by gray dots. Values represent mean ± SEM. (ns, non-significant difference). DG, dentate gyrus; gcl, granule cell layer; ml, molecular layer.

changes in synaptic strength and intrinsic cellular properties detected 6 hr after atRA injections prompted the hypothesis, that atRA may modulate the ability of neurons to express synaptic plasticity.

To test the effects of atRA on the ability of neurons to express synaptic plasticity, long-term potentiation (LTP) experiments on perforant path synapses to dentate granule cells were carried out in the dorsal hippocampus of anesthetized mice (*Figure 7*). atRA (10 mg/kg) was injected intraperitoneally 3–6 hr prior to experimental assessment and LTP was probed through electric stimulation of the perforant path with a theta-burst stimulation (TBS) protocol (*Figure 7A*). Consistent with our single-cell recordings, which showed no major differences in sEPSC amplitudes and intrinsic cellular properties of dentate granule cells in the atRA group (c.f., *Figures 1–4*), no significant difference

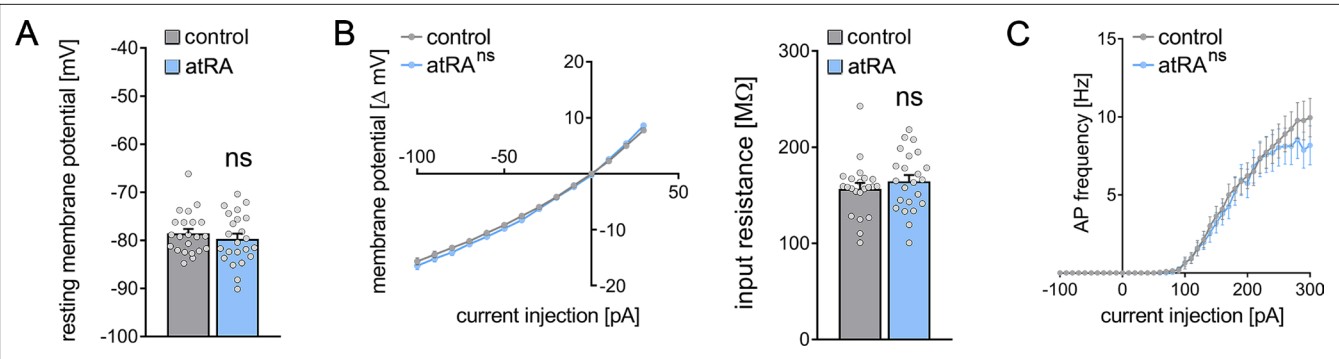

**Figure 4.** Passive or active membrane properties of dentate granule cells remain unchanged in the ventral hippocampus following intraperitoneal administration of all-trans retinoic acid (atRA). (**A–C**) Group data of resting membrane potentials (**A**), input-output curves, and input resistances (**B**), and action potential (AP) frequency of dentate granule cells in the ventral hippocampus ($n_{control}$=22 cells, $n_{atRA}$=23 cells in four animals each; Mann-Whitney test for column statistics, RM two-way ANOVA followed by Sidak's multiple comparisons test for input-output curve and AP frequency analysis). Individual data points are indicated by gray dots. Values represent mean ± SEM. (ns, non-significant difference).

in input-output properties was observed between atRA-treated and vehicle-only animals in these experiments (*Figure 7B*). Weak TBS was applied to the medial perforant path. As shown in *Figure 7C*, a significant increase in the fEPSP slopes was observed in both groups upon plasticity induction. However, the increased fEPSP slopes persisted for at least 60 min in the atRA-treated mice. We concluded that atRA promotes the ability of dentate granule cells to express long-term synaptic plasticity while not critically affecting baseline synaptic transmission before plasticity induction.

To confirm and extend these findings, LTP experiments were carried out in synaptopodin-deficient mice. We recently demonstrated that synaptopodin is required for atRA-mediated synaptic strengthening in the neocortex (*Lenz et al., 2021*). Indeed, atRA- and vehicle-only-treated synaptopodin-deficient animals were indistinguishable in these experiments, and within 60 min, fEPSP slopes had returned to baseline in both groups (*Figure 7B and D*). Taken together, we concluded that the intraperitoneal administration of atRA induces metaplastic changes and that the presence of synaptopodin is required for atRA-mediated metaplasticity.

## Discussion

In this study, we investigated the effects of systemic, that is, the intraperitoneal application of atRA on synaptic transmission and the plasticity of mature dentate granule cells in the adult hippocampus. Our results demonstrate that atRA promotes the ability of dentate granule cells to express synaptic plasticity. Specifically, a persistent strengthening of excitatory neurotransmission was observed after in vivo LTP-induction in the atRA-treated animals. In line with our recent findings, we showed that the presence of synaptopodin is required for atRA-mediated synaptic plasticity (*Lenz et al., 2021*). Aside from increased sEPSC frequencies and synapse numbers in the dorsal hippocampus, atRA had no significant effect on baseline excitatory and inhibitory synaptic strength in the dentate gyrus. Hence, we propose that atRA modulates the ability of neurons to express synaptic plasticity consistent with a synaptopodin-dependent metaplastic effect of atRA.

Vitamin A metabolites, such as atRA and their related signaling pathways, have been linked to various physiological brain functions, including synaptic plasticity (*Chen et al., 2014*; *Lenz et al., 2021*; *Arendt et al., 2015*; *Zhong et al., 2018*). Accordingly, atRA has been evaluated in disease models and patients with brain disorders associated with cognitive decline, including Alzheimer's disease, Fragile X syndrome, and depression (*Zhang et al., 2018*; *Bremner et al., 2012*; *Ding et al., 2008*; *Soden and Chen, 2010*; *Park et al., 2021*). However, the precise mechanisms through which atRA asserts its effects on synaptic transmission and plasticity in health and disease are subjects of further investigations.

In a recent study, we demonstrated that atRA induces structural and functional synaptic changes in neurons of the adult human cortex (*Lenz et al., 2021*). Specifically, an increase in the sEPSC amplitudes was observed 6 hr after exposure of acute cortical slices to 1 µM atRA. These functional changes were highly correlated with increased spine head sizes, and corresponding changes in synaptopodin

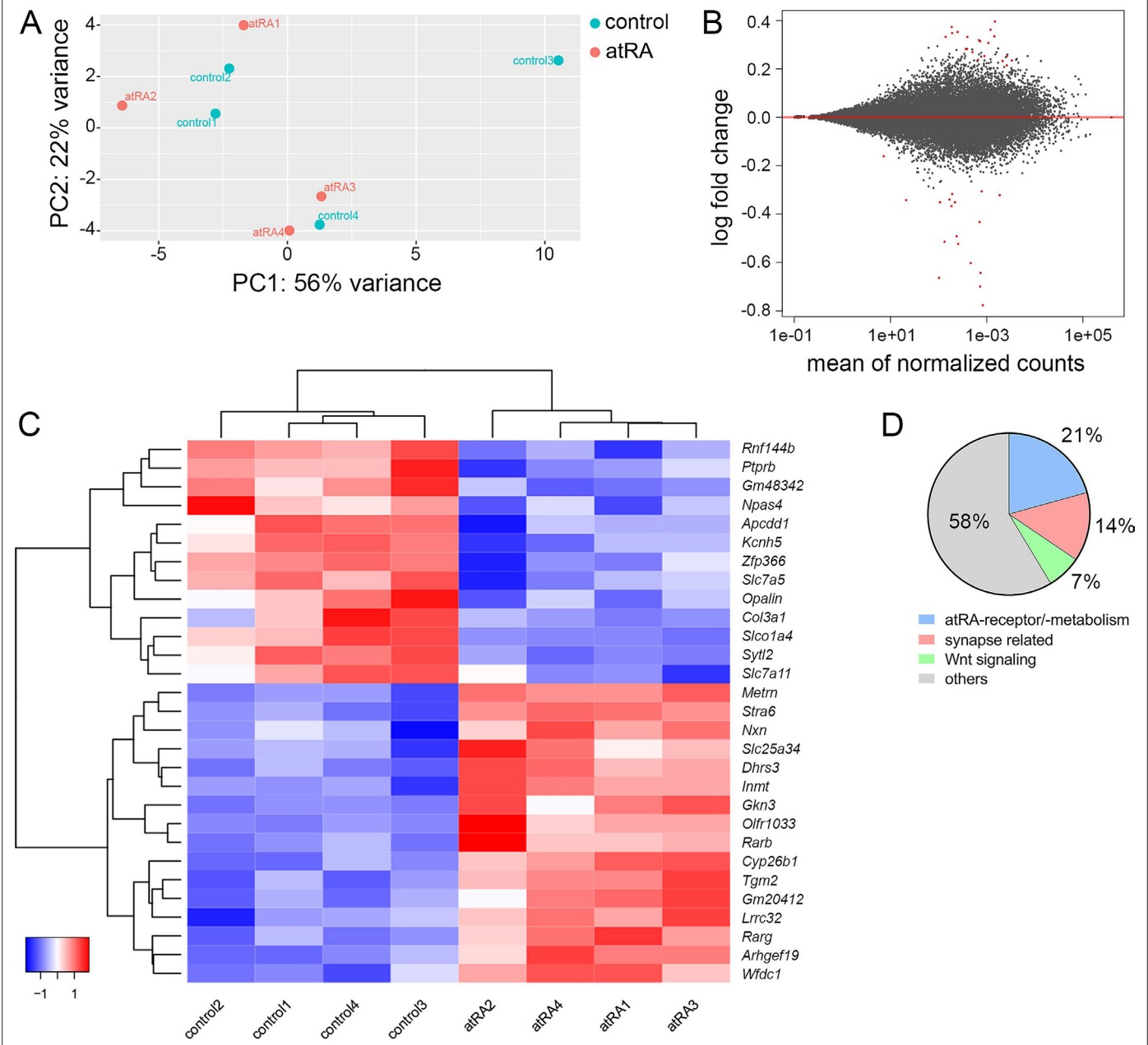

**Figure 5.** Hippocampal transcriptome analysis reveals no major differences in synapse-related genes following intraperitoneal injection of all-trans retinoic acid (atRA). (**A**) Principal component analysis with 'treatment' as primary factor reveals no treatment-specific clustering of hippocampal mRNA samples (n=4 animals, one hippocampus each). (**B**) DESeq2-Analysis reveals differential expression of genes with a moderate |log2FC|<1 (visualization by MA plot). (**C**) Heatmap showing the z-scores of 29 differentially expressed genes. The differential expression of genes depends on the atRA treatment, as indicated by the z-score clustering. (**D**) Subsets of genes can be attributed to atRA-signaling or atRA-metabolism, synaptic transmission, and Wnt-signaling, respectively.

clusters and spine apparatus organelles were observed (*Lenz et al., 2021*). Consistent with these findings, increased sEPSC amplitudes were observed in acute cortical slices prepared from the medial prefrontal cortex of wild-type but not synaptopodin-deficient mice (*Lenz et al., 2021*). These findings identified atRA as a potent mediator of synaptic plasticity in the adult human cortex. Furthermore, they suggest that synaptopodin-dependent signaling pathways are involved in mediating the synaptic effects of atRA.

In the present study, however, we did not observe any changes in excitatory synaptic strength in dentate granule cells in either the ventral or the dorsal hippocampus. Specifically, no changes in

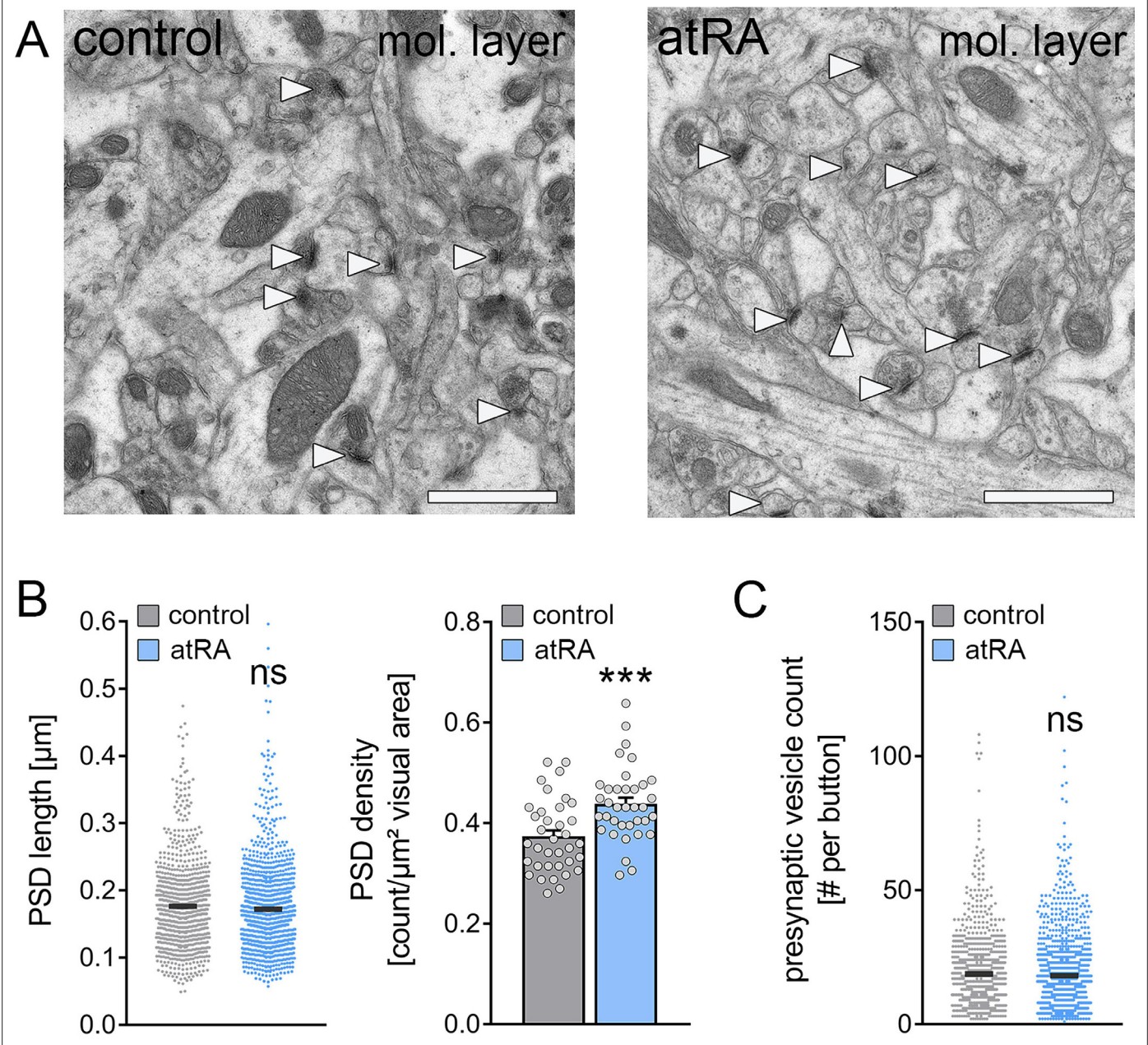

**Figure 6.** The numbers of cortico-hippocampal synapses in the dentate gyrus of the dorsal hippocampus are increased following intraperitoneal administration of all-trans retinoic acid (atRA). (**A**) Representative electron microscopy images of asymmetric synapses (arrowheads) in the outer two-thirds of the molecular layer (mol. layer) of the dorsal hippocampus from atRA- or vehicle-only-injected mice. Scale bar=1 μm. (**B**) Group data of postsynaptic density (PSD) counts ($n_{control}$=36 individual images, $n_{atRA}$=36 individual images in three different animals with 12 images per sample; Mann-Whitney test, U=340) and analysis of PSD lengths of asymmetric synapses ($n_{control}$=908 PSDs, $n_{atRA}$=1189 PSDs in three different animals, one data point outside the axis limits in the control group; Mann-Whitney test). (**C**) Presynaptic vesicle counts are not significantly different between the groups ($n_{control}$=885 presynaptic buttons, $n_{atRA}$=1151 presynaptic buttons in three different animals, one data point outside the axis limits in the atRA group; Mann-Whitney test). Individual data points are indicated by gray dots. Values represent mean ± SEM. (\*\*\*, p<0.001; ns, non-significant difference).

the sEPSC amplitudes were observed (*Lenz et al., 2021*). Consistent with these results no major changes in the ultrastructural properties of excitatory synapses, that is PSD lengths and presynaptic vesicle counts of asymmetric synapses, were observed between the two groups. Interestingly, ultra-structural analysis revealed an increase in the number of asymmetric synapses in the dorsal hippo-campus of atRA-treated animals. Previous studies reported differential effects of atRA in distinct brain regions. For example, in the visual cortex, a reduction in inhibitory neurotransmission and no effect on

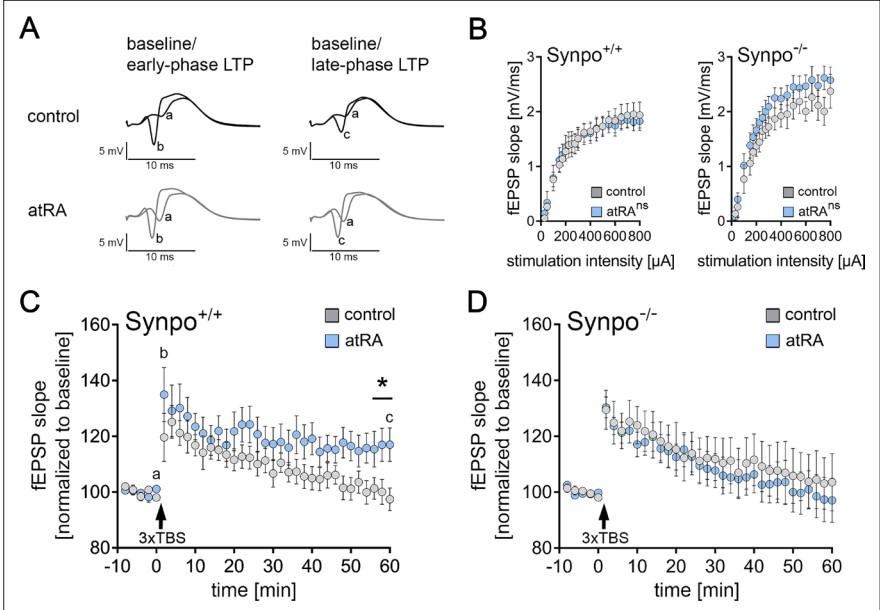

**Figure 7.** Intraperitoneal injection of all-trans retinoic acid (atRA) improves synaptic plasticity in the dentate gyrus of wild-type but not synaptopodin-deficient mice. (**A**) In vivo long-term potentiation (LTP) experiments on perforant path synapses were carried out in anesthetized mice using a weak theta-burst stimulation (TBS) protocol. Representative traces of field excitatory postsynaptic potential (fEPSP) recordings in wild-type mice at indicated points in time (a–c) after induction of LTP in vehicle-only controls and atRA-injected mice (10 mg/kg, i.p.; 3–6 hr prior to recordings). (**B**) Input-output properties of wild-type and synaptopodin-deficient animals (Synpo$^{+/+}$: $n_{control}$=9 animals, $n_{atRA}$=9 animals; Synpo$^{-/-}$: $n_{control}$=7 animals, $n_{atRA}$=8 animals. RM two-way ANOVA with Sidak's multiple comparisons). (**C**) Group data of fEPSP slopes in wild-type mice (Synpo$^{+/+}$: $n_{control}$=9 animals, $n_{atRA}$=9 animals; Mann-Whitney test, U=13–17 for three terminal data points). (**D**) Group data of fEPSP slopes in synaptopodin-deficient mice (Synpo$^{-/-}$: $n_{control}$=7 animals, $n_{atRA}$=8 animals; Mann-Whitney test). Values represent mean ± SEM. (*, $p<0.05$; ns, non-significant difference).

excitatory neurotransmission were observed (*Zhong et al., 2018*), while in the somatosensory cortex, evidence for increased inhibitory synaptic strength in the absence of changes in excitatory neurotransmission was provided (*Yee and Chen, 2016*). Likewise, in the hippocampal CA1 region, atRA seems to potentiate excitatory synapses while depressing inhibitory synapses (*Sarti et al., 2013*). These findings support the notion that atRA may assert its effect on synaptic transmission in a brain region- or cell type-specific manner. However, differences in the respective experimental settings must be carefully considered, such as the use of distinct tissue preparations (acute slices vs. organotypic tissue cultures vs. dissociated neurons) and differences in atRA administration (systemic vs. local; in vivo vs. ex vivo). Consequently, additional research is required to better understand the distinct effects of atRA on synaptic transmission and to determine, for example, possible concentration-dependent effects, the impact of a single dose versus repeated (long-term) administration, and the cell-type specific responsiveness to atRA. Nevertheless, the results of the present study demonstrate increased sEPSC frequencies and synapse numbers in the dorsal hippocampus of atRA-treated animals, thereby confirming that atRA targets excitatory synapses in the dorsal hippocampus.

The results of our mRNA analysis showed that intraperitoneally injected atRA affects the hippocampus and leads to transcriptional changes related to atRA-signaling/atRA-metabolism. These findings concur with previous work demonstrating RA signaling in the hippocampus (*Goodman et al., 2012*). Notably, although a small number of genes showed transcriptional alterations, we did not observe any major changes in synaptic or plasticity-related genes. Thus, it appears unlikely that atRA affects plasticity primarily by regulating the transcription of plasticity-related genes. Rather, as has been suggested by others, atRA-induced changes in mRNA translation can account for the rapid effects of atRA on synaptic plasticity (*Poon and Chen, 2008*; *Maghsoodi et al., 2008*). In line with this interpretation, our previous work revealed that atRA-mediated synaptic changes are not observed when protein synthesis is blocked with anisomycin in mouse and human neocortical slices (*Lenz et al.,*

*2021*), suggesting that atRA modulates protein synthesis and, therefore, the availability of proteins required for the induction of synaptic plasticity. Although these findings do not fully exclude the possibility that atRA-related transcriptional changes influence synaptic plasticity to some extent (*Garay et al., 2020*), our data are in line with the effects of atRA on local protein synthesis, which plays a major role in different forms of synaptic plasticity, including metaplasticity (*Kang and Schuman, 1996*; *Sutton et al., 2006*; *Woo and Nguyen, 2003*).

It is interesting to theorize that atRA may act as a permissive rather than an instructive plasticity factor in this context. That is to say, atRA may not induce specific changes in excitatory and inhibitory neurotransmission in distinct brain regions but rather act by influencing the ability of neurons to express structural and functional synaptic plasticity. Thus, the specific outcome of atRA treatment on excitatory and inhibitory neurotransmission may depend on the specific stimuli applied or changes in network activity occurring after atRA administration. Since we did not observe major changes in baseline synaptic transmission, aside from changes in the sEPSC frequencies in the dorsal hippocampus and because no significant changes in input-output properties were observed in our in vivo electrophysiological recordings, we were able to test for such permissive, metaplastic effects of atRA. Therefore, in vivo LTP was probed with a mild plasticity-inducing stimulus (*Jedlicka et al., 2015*). In the absence of any differences in input-output properties before LTP-induction, atRA promoted the ability of neurons to maintain increased synaptic strength 60 min after LTP-induction. These findings are consistent with atRA-mediated metaplasticity.

The plasticity-promoting effects of atRA were not observed in synaptopodin-deficient mice, suggesting that synaptopodin is required for atRA-mediated metaplasticity. These findings are in line with our previous work, which showed that the presence of synaptopodin is required for atRA-mediated synaptic strengthening to occur in the mouse prefrontal cortex (*Lenz et al., 2021*). Moreover, we were able to demonstrate that atRA triggers an increase in synaptopodin clusters and spine apparatus sizes in human cortical slices (*Lenz et al., 2021*). These findings call for a systematic assessment of atRA-mediated synaptopodin-dependent synaptic plasticity, including assessing specific stimuli, network states, and other conditions that may trigger associative and homeostatic changes in excitatory and inhibitory neurotransmission. Whether and how ultrastructural changes of spine apparatus organelles (*Lenz et al., 2021*; *Vlachos et al., 2013*; *Galanis et al., 2021*) reflect the induction of different forms of synaptic plasticity is currently unknown. Considering that both atRA-signaling and synaptopodin-mediated signaling pathways have been associated to pathological brain states, such as Alzheimer's disease (*Endres, 2019*; *Fahrenholz et al., 2010*; *Goetzl et al., 2016*; *Aloni et al., 2019*; *Wingo et al., 2019*), we are confident that a better understanding of atRA-mediated synaptopodin-dependent synaptic plasticity may support the development of novel therapeutic strategies aimed at synaptic plasticity modulation.

# Materials and methods

**Key resources table**

| Reagent type (species) or resource | Designation | Source or reference | Identifiers | Additional information |
|---|---|---|---|---|
| Chemical compound, drug | All-trans retinoic acid (atRA) | Sigma-Aldrich | Cat#: R2625 | Final concentration: 10 mg/kg Injection vehicle: Corn oil + 5% DMSO |
| Chemical compound, drug | Dimethyl Sulfoxide (DMSO) | Sigma-Aldrich | Cat#: D2650 | |
| Chemical compound, drug | Paraformaldehyde (PFA) | Carl Roth | Cat#: 0335.3 | Final concentration: 4% (w/v) in PB or PBS |
| Chemical compound, drug | Glutardialdehyd | Carl Roth | Cat#: 4157.2 | Final concentration: 2% (w/v) in PB |
| Chemical compound, drug | CNQX | Biotrend | Cat#: BN0153 | Final concentration: 10 µM |
| Chemical compound, drug | D-APV | Abcam | Cat#: ab120003 | Final concentration: 10 µM |
| Chemical compound, drug | DAPI (1 mg/ml in water) | Thermo Fisher Scientific | Cat#: 62248 | IF and post hoc labeling (1:5000) |

*Continued on next page*

*Continued*

| Reagent type (species) or resource | Designation | Source or reference | Identifiers | Additional information |
|---|---|---|---|---|
| Commercial assay or kit | Monarch Total RNA Miniprep Kit | New England Biolabs | #T2010S | |
| Genetic reagent (*Mus musculus*) | B6.129-Synpo^tm1Mndl/Dllr; Synpo^−/− | *Vlachos et al., 2013* PMID:23630268 | RRID:MGI:6423115 | Obtained from Deller Lab (Frankfurt); male |
| Peptide, recombinant protein | Streptavidin, Alexa Fluor 488-Conjugate | Invitrogen | Cat#: S32354 RRID:AB_2315383 | Post hoc labeling (1:1000) |
| Software, algorithm | Prism | GraphPad | RRID:SCR_002798 | |
| Software, algorithm | Clampfit (pClamp software package) | Molecular Devices | RRID:SCR_011323 | |
| Software, algorithm | ImageJ | | RRID:SCR_003070 | |
| Software, algorithm | Photoshop | Adobe | RRID:SCR_014199 | |
| Strain, strain background (*M. musculus*) | C57BL/ 6J; Synpo^+/+ | Jackson Laboratory | RRID:IMSR_JAX:000664 | |

## Pharmacological treatment

AtRA (Sigma-Aldrich) was dissolved in DMSO and stored at − 20°C until further use. The injection solution was prepared immediately before injection by adding corn oil to prediluted stocks to achieve a final concentration of 5% DMSO (v/v). Before use, the solution was vortexed briefly. The solution was intraperitoneally injected in adult (C57BL//6J; 6–10 weeks old) male mice at an atRA concentration of 10 mg/kg. Control animals were injected with a vehicle-only solution (5% DMSO in corn oil) but otherwise treated equally. After injection, no overt behavioral changes were observed. Experiments were performed 3–6 hr after intraperitoneal injections.

## Preparation of acute mouse hippocampal slices

Adult mice were anesthetized with ketamine/xylazine (100 mg/kg ketamine and 20 mg/kg xylazine) and rapidly decapitated. Brains were removed and further dissected for the preparation of acute slices of the ventral hippocampus as previously described (*Bischofberger et al., 2006*). For the preparation of acute slices of the dorsal hippocampus, the rostral and caudal parts of the brains were removed to ensure stable coronal sectioning. Brains were immediately transferred to a cooled oxygenated extracellular solution ( 5°C; 5% $CO_2$ / 95% $O_2$) containing (in mM): 92 NMDG, 2.5 KCl, 1.25 $NaH_2PO_4$, 30 $NaHCO_3$, 20 HEPES, 25 glucose, 2 thiourea, 5 Na-ascorbate, 3 Na-pyruvate, 0.5 $CaCl_2$, and 10 $MgSO_4$; pH = 7.3–7.4 at ~ 7°C (NMDG-aCSF; *Ting et al., 2018*). 300 µm tissue sections were cut with a Leica VT1200S vibratome. Slices were transferred to cell strainers with 40 µm pore size placed in NMDG-aCSF at 34°C, and the sodium levels were gradually increased following a protocol as described before (*Ting et al., 2018*). After recovery, slices were maintained for further experimental assessment at room temperature in extracellular solution containing (in mM): 92 NaCl, 2.5 KCl, 1.25 $NaH_2PO_4$, 30 $NaHCO_3$, 20 HEPES, 25 glucose, 2 thiourea, 5 Na-ascorbate, 3 Na-pyruvate, 2 $CaCl_2$, and 2 $MgSO_4$.

## Whole-cell patch-clamp recordings

Dentate granule cells in the suprapyramidal blade of the dentate gyrus were recorded in a bath solution ( 35°C) containing (in mM): 92 NaCl, 2.5 KCl, 1.25 $NaH_2PO_4$, 30 $NaHCO_3$, 20 HEPES, 25 glucose, 2 thiourea, 5 Na-ascorbate, 3 Na-pyruvate, 2 $CaCl_2$, and 2 $MgSO_4$. Granule cell somata close to the molecular layer of the dentate gyrus were visually identified using an LN-Scope (Luigs & Neumann, Ratingen, Germany) equipped with an infrared dot-contrast and a 40× water immersion objective (Olympus, NA 0.8). Recorded signals were amplified using a Multiclamp 700B amplifier, digitized with a Digidata 1550B digitizer and visualized with the pClamp 11 software package. For recordings of sEPSC and intrinsic cellular properties, patch pipettes with a tip resistance of 3–5 MΩ were used, containing (in mM): 126 K-Gluconate, 4 KCl, 10 HEPES, 4 MgATP, 0.3 $Na_2GTP$, 10 PO-Creatine, and 0.3 % (w/v) Biocytin (pH=7.25 with KOH, 285 mOsm/kg). For sEPSC recordings, dentate granule cells were held at –80 mV in voltage-clamp mode. Intrinsic cellular properties were recorded in current-clamp mode. A pipette capacitance of 2.0 pF was corrected and

series resistance was compensated using the automated bridge balance tool of the Multiclamp commander. IV curves were generated by injecting 1 s square pulse currents starting at –100 pA and increasing in 10 pA steps until +500 pA injection was reached (sweep duration: 2 s). sIPSCs were recorded in the same extracellular solution by adding the AMPA receptor inhibitor CNQX (10 μM, Biotrend) and the NMDA receptor inhibitor APV (10 μM, Abcam). Patch pipettes for sIPSC recordings contained (in mM): 40 CsCl, 90 K-gluconate, 1.8 NaCl, 1.7 MgCl$_2$, 3.5 KCl, 0.05 EGTA, 2 MgATP, 0.4 Na$_2$GTP, 10 PO-Creatine, and 10 HEPES (pH=7.25 with KOH, 290 mOsm), and granule cells were held at –70 mV during the recordings. Series resistance was monitored and recordings were discarded if series resistance reached >30 MΩ.

## Post hoc labeling of patched dentate granule cells

Acute slice preparations were fixed in 4% PFA/ 4% sucrose (w/v, phosphate-buffered saline, PBS) at room temperature and stored at 4°C overnight in the same solution. After fixation, slices were washed in PBS and consecutively incubated for 1 hr with 10% (v/v) normal goat serum (NGS) in 0.5% (v/v) Triton X-100 containing PBS to reduce unspecific staining. For post hoc visualization of patched dentate granule cells, sections were incubated for 3 hr with streptavidin-Alexa Fluor 488 (Invitrogen, #S32354; 1:1000 dilution in 10% (v/v) NGS, 0.1% (v/v) Triton X-100 containing PBS) at room temperature. Sections were washed in PBS and incubated with DAPI for 10 min (Thermo Fisher Scientific, #62248; 1:5000 dilution in PBS) to visualize the cytoarchitecture. After washing, sections were transferred onto glass slides and mounted with fluorescence anti-fading mounting medium (DAKO Fluoromount). Confocal images were acquired using a Leica SP8 laser-scanning microscope equipped with a 20× multi-immersion (NA 0.75; Leica) and a 40× oil-immersion (NA 1.30; Leica) objective. Image stacks were acquired in tile scanning mode with the automated stitching function of the LasX software package.

## Electron microscopy

Adult mice of both sexes were anesthetized using intraperitoneal injection of ketamine (100 mg/kg) and xylazine (20 mg/kg). Deeply anesthetized mice were transcardially perfused using 2% (w/v; 0.1 M phosphate buffer, PB) glutaraldehyde and 4% (w/v; 0.1 M PB) paraformaldehyde. Post hoc fixation of the brains was continued overnight in the same fixation solution. After fixation, frontal sections containing the dorsal hippocampus were generated using a Leica VT1000S vibratome. Isolated dorsal hippocampal slices were washed for 4 hr in 0.1 M PB. Subsequently, slices were incubated with 1% osmium tetroxide for 60 min, washed in graded ethanol (up to 50% (v/v)) for 5 min each, and incubated overnight with uranyl acetate ( 1% (w/v) in 70% (v/v) ethanol) overnight. Slices were then dehydrated in graded ethanol (80%, 90%, and 98% for 5 min each, 2 times 100 % for 10 min each). Subsequently, two washing steps were performed in propylene oxide for 10 min each prior to incubation with durcupan/propylene oxide (1:1 for 1 hr) and transferred to durcupan (overnight at room temperature). Slices were embedded in durcupan, and ultra-thin sectioning (55 nm) was performed using a Leica UC6 Ultracut. Sections were mounted onto copper grids (Plano), at which point an additional Pb-citrate contrasting step was performed (3 min). Electron microscopy was performed with a LEO 906E microscope (Carl Zeiss) at 4646× magnification. For each sample, 12 images from the outer two-thirds of the molecular layer were acquired and further analyzed.

## RNA isolation and transcriptome analysis

Hippocampi were isolated from the brain of adult mice and immediately transferred into RNA protection buffer (New England Biolabs) and RNA was consecutively isolated using a column-based RNA isolation kit according to the manufacturer's instructions (Monarch Total RNA Miniprep Kit; #T2010S New England Biolabs). Strand-specific cDNA library preparation from polyA enriched RNA (150 bp mean read length) and RNA sequencing was performed by Eurofins Genomics (Eurofins Genomics Europe Sequencing GmbH, Konstanz, Germany). RNA sequencing was performed using the genome sequencer Illumina HiSeq technology in NovaSeq 6000 S4 PE150 XP sequencing mode. For further analysis .fastq-files were provided. All files contained more than 45 M high-quality reads having at least a phred quality of 30 (> 90% of total reads).

## In vivo perforant path long-term potentiation

Three-month-old C57BL/ 6J (Synpo$^{+/+}$) or synaptopodin-deficient male animals (Synpo$^{-/-}$; with C57BL/6J genetic background) were kept in a 12 hr light/12 hr dark cycle (Scantainer) with access to food and water ad libitum. To achieve stable anesthesia, an initial dose of urethane (1.25 g/kg, in sodium chloride solution) was injected subcutaneously (s.c.); a supplemental dose of 0.1 g/kg was given as needed. After stable anesthesia was reached, atRA (10 mg/kg in 5% DMSO) or vehicle-only was intraperitoneally injected (blind to experimenter). The surgery and electrode placement were performed as previously described (*Jedlicka et al., 2011*; *Muellerleile et al., 2020*). Briefly, the mouse was placed in a stereotactic frame (David Kopf Instruments) and local anesthesia with prilocaine (Xylonest 1%, Astra Zeneca, s.c. to the scalp) was applied. Cranial access to the brain was established according to coordinates from the mouse brain atlas (Franklin and Paxinos; stimulation electrode: 2.5 mm lateral to the midline, 3.8 mm posterior to bregma; recording electrode: 1.2 mm lateral to the midline, 1.7 mm posterior to bregma). The ground electrode was placed in the neck musculature. Electrophysiological signals were amplified using a Grass P55 A.C. pre-amplifier (Astro-Med) and digitized at a 10 kHz sampling rate (Digidata 1440 A, Molecular Devices). Extracellular stimulation was performed using a STG1004 stimulator (Multichannel Systems). A bipolar stimulation electrode (NE-200, 0.5 mm tip separation, Rhodes Medical Instruments) was lowered 1.5–2.2 mm below the surface of the brain to target the angular bundle of the perforant path. Then a tungsten recording electrode (TM33B01KT, World Precision Instruments) was lowered in 0.1 mm increments while monitoring the waveform of the field excitatory postsynaptic potential (fEPSP) in response to 500 μA test pulses until the granule cell layer in the dorsal part of the hippocampus was reached (1.7–2.2 mm below the surface). The correct placement of the stimulation electrode in the medial portion of the perforant path was verified electrophysiologically by the latency of the population spike (approximately 4 ms), although the activation of some lateral perforant path fibers could not be excluded. Recordings started a minimum of 3 hr after experimental treatment with atRA or vehicle-only. An input-output curve was generated by 30–800 μA current pulses, repeated three times at each intensity, 0.1 ms pulse duration, 60 pulses total at 0.1 Hz. Perforant path-dentate gyrus (PP/DG)-LTP was recorded by applying stimuli with a current intensity set to elicit a 1–2 mV population spike (0.1 Hz, 0.1 ms pulse duration). PP/DG-LTP was induced using a weak TBS protocol (*Jedlicka et al., 2015*) composed of three series of six trains with six 400 Hz current pulses at double the baseline intensity and pulse duration (with 200 ms interval between trains and 20 s interval between series). Following LTP induction, evoked fEPSPs were recorded for 1 hr using the baseline stimulation parameters.

## Quantification and statistics

RNA sequencing data were uploaded to the galaxy web platform (public server: usegalaxy.eu; *Afgan et al., 2018*; *Jalili et al., 2020*; *Afgan et al., 2016*) and transcriptome analysis was performed using the Galaxy platform in accordance with the reference-based RNA-seq data analysis tutorial (*Batut et al., 2018*). Adapter sequences, low quality, and short reads were removed via the CUTADAPT tool (Galaxy version 1.16.5). Reads were mapped using RNA STAR (Galaxy version 2.7.6 a) with the mm10 Full reference genome (*Mus musculus*). For an initial assessment of gene expression, unstranded FEATURECOUNT (Galaxy version 2.0.1) analysis was performed from RNA STAR output. Statistical evaluation was performed using DESeq2 (Galaxy version 2.11.40.6+ galaxy1) with treatment as the primary factor that might affect gene expression. Genes were considered as differentially expressed if the adjusted p-value was <0.05. Heatmaps were generated based on z-scores of the normalized count table.

Single-cell recordings were analyzed off-line using Clampfit 11 of the pClamp11 software package (Molecular Devices). sEPSC and sIPSC properties were analyzed using the automated template search tool for event detection (*Lenz et al., 2021*). Input resistance was calculated for the injection of –100 pA current at a time frame of 200 ms with a maximum distance to the initial hyperpolarization. Resting membrane potential was calculated as the mean baseline value. AP detection was performed using the input-output curve threshold search event detection tool, and the AP frequency was assessed by the number of APs detected during the respective current injection time. One individual cell (control group, ventral hippocampus) was excluded from the analysis of intrinsic membrane properties, since the membrane patch lost its integrity during the recordings. The AP plots in *Figures 2 and 4* depict cellular responses until 300 pA current injection. Beyond 300 pA, subsets of granule cells in both

groups failed to maintain regular AP firing. The treatment did not significantly affect AP frequency beyond 300 pA current injection. In vivo perforant path LTP was analyzed using Clampfit 10.2 and custom MATLAB (Mathworks) scripts. In these experiments, one Synpo$^{-/-}$ animal in the vehicle-only group was excluded from further analysis, since an insufficient response to increasing stimulus intensities was detected in the input-output curve.

Electron microscopy images were analyzed and cross-checked by five investigators blind to experimental conditions. Image analysis was performed using the ImageJ software package (available at http://imagej.nih.gov/ij/). For PSD assessment, all visible PSDs in one image were counted and normalized to the image area. Subsequently, PSD length and presynaptic vesicle abundance was manually assessed. In case of perforated synapses, each PSD was analyzed individually. For statistical evaluation, each individual synapse was considered a biological replicate.

Data were statistically evaluated using GraphPad Prism 7 (GraphPad Software, USA). Statistical comparisons were made using the nonparametric Mann-Whitney test. For statistical comparison of XY plots in whole-cell patch-clamp recordings and fEPSP input-output curves, we used an RM two-way ANOVA test (repeated measurements/analysis) with Sidak's multiple comparisons. Statistical analysis of fEPSP slope data was performed using the Mann-Whitney test for the three terminal data points. p-values smaller than 0.05 were considered a significant difference. In the text and figures, values represent mean ± standard error of the mean (SEM). Statistical significance in XY plots is indicated in the figure panel. U-values were provided for significant results only. *, p<0.05; ***, p<0.001; ns, non-significant differences.

## Digital illustrations

Figures were prepared using Photoshop graphics software (Adobe, San Jose, CA). Image brightness and contrast were adjusted.

## Acknowledgements

The authors would like to thank Sigrun Nestel and Simone Zenker for their technical assistance. The work was supported by Else Kröner-Fresenius-Stiftung (EKFS_#2019_A94 to ML) and Deutsche Forschungsgemeinschaft (DFG; CRC 1080 to TD and AV; Project ID 259373024 B14 - CRC/TRR 167 to AV). The Galaxy server that was used for some calculations in this study is partially funded by Collaborative Research Centre 992 Medical Epigenetics (DFG grant SFB 992/1 2012) and German Federal Ministry of Education and Research (BMBF grants 031 A538A/A538C RBC, 031L0101B/031L0101C de.NBI-epi, and 031L0106 de.STAIR (de.NBI)).

## Additional information

### Competing interests

Thomas Deller: received funding from Novartis for a lecture on human brain anatomy. The other authors declare that no competing interests exist.

### Funding

| Funder | Grant reference number | Author |
| --- | --- | --- |
| Else Kröner-Fresenius-Stiftung | EKFS_#2019_A94 | Maximilian Lenz |
| Deutsche Forschungsgemeinschaft | CRC 1080 | Thomas Deller Andreas Vlachos |
| Deutsche Forschungsgemeinschaft | Project-ID 259373024 B14 - CRC/TRR 167 | Andreas Vlachos |

The funders had no role in study design, data collection and interpretation, or the decision to submit the work for publication.

## Author contributions
Maximilian Lenz, Conceptualization, Formal analysis, Funding acquisition, Investigation, Project administration, Validation, Visualization, Writing - original draft; Amelie Eichler, Pia Kruse, Julia Muellerleile, Formal analysis, Investigation; Thomas Deller, Funding acquisition, Resources; Peter Jedlicka, Formal analysis, Investigation, Resources; Andreas Vlachos, Conceptualization, Funding acquisition, Methodology, Project administration, Resources, Supervision, Writing - original draft

## Author ORCIDs
Maximilian Lenz (iD) http://orcid.org/0000-0003-3147-4949
Amelie Eichler (iD) http://orcid.org/0000-0001-7990-654X
Pia Kruse (iD) http://orcid.org/0000-0002-1742-1608
Julia Muellerleile (iD) http://orcid.org/0000-0001-9249-5749
Thomas Deller (iD) http://orcid.org/0000-0002-3931-2947
Andreas Vlachos (iD) http://orcid.org/0000-0002-2646-3770

## Ethics
Ethics statementAll experiments were performed according to German animal welfare legislation and after positive evaluation by the local authorities (University of Freiburg, AZ G-19/152; Faculty of Medicine at the University of Frankfurt, AZ FU/1131). Animals were kept in a 12 hour light/12 hour dark cycle with access to food and water ad libitum. Every effort was made to minimize pain or distress of animals.

## Decision letter and Author response
Decision letter https://doi.org/10.7554/eLife.71983.sa1
Author response https://doi.org/10.7554/eLife.71983.sa2

# Additional files

## Supplementary files
• Transparent reporting form

## Data availability
Data and statistical analysis (Software: GraphPad Prism) are accessible through the following link (Dryad Digital Repository): https://doi.org/10.5061/dryad.5qfttdz5t. RNA sequencing data are accessible from the Galaxy web platform via the following link: https://usegalaxy.eu/u/maximilian.lenz/h/transcriptome-analysisatra-6h-vs-controlhippocampus. Custom MATLAB scripts for fEPSP population spike analysis are accessible through the following link (https://github.com/juliamuellerleile/population-spike-analysis, copy archived at https://archive.softwareheritage.org/swh:1:rev:f3b486129684a2e5218550ff370c5abb36509967).

The following dataset was generated:

| Author(s) | Year | Dataset title | Dataset URL | Database and Identifier |
|---|---|---|---|---|
| Lenz M, Eichler A, Kruse P, Muellerleile J, Deller T, Jedlicka P, Vlachos A | 2021 | All-trans retinoic acid induces synaptopodin-dependent metaplasticity in mouse dentate granule cells | https://doi.org/10.5061/dryad.5qfttdz5t | Dryad Digital Repository, 10.5061/dryad.5qfttdz5t |
| Lenz M, Eichler A, Kruse P, Muellerleile J, Deller T, Jedlicka P, Vlachos A | 2021 | All-trans retinoic acid induces synaptopodin-dependent metaplasticity in mouse dentate granule cells | https://usegalaxy.eu/u/maximilian.lenz/h/transcriptome-analysisatra-6h-vs-controlhippocampus | Galaxy Europe, transcriptome-analysisatra-6h-vs-controlhippocampus |

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
