## [Decision Letter]

**Decision letter after peer review:**

Thank you for submitting your article "All-trans retinoic acid induces synaptopodin-dependent metaplasticity in mouse dentate granule cells" for consideration by *eLife*. Your article has been reviewed by 2 peer reviewers, and the evaluation has been overseen by Lu Chen as the Senior and Reviewing Editor. The following individual involved in review of your submission has agreed to reveal their identity: Michael A Sutton (Reviewer #2).

Essential revisions:

1) The authors claim that atRA does not have major effects on excitatory synaptic transmission in the DG neurons of hippocampus. However, the reviewers noticed that sEPSC frequency was increased significantly in the dorsal DG neurons. While we do not think more experiments are necessary, this change should not be ignored. Please provide some plausible explanations to this observation.

2) Please specify the part of DG analyzed in in vivo study as dorsal and ventral DG neurons responded differently in slice physiology.

3) Compared to cortical neurons (from your previous publication) and many other studies on hippocampal pyramidal neurons, hippocampal DG neurons seem to be more resistant to the synapse-enhancing effects of atRA. While more experiments would help resolve whether granule cells are truly resistant to atRA's effects on basal synaptic function (or whether this is a route of administration/concentration issue), we think the primary conclusion is well supported by data presented. Thus, no additional experiments are required at this point, but the authors may want to make this point more clearly in the discussion.

*Reviewer #1 (Recommendations for the authors):*

The author needs to, at a minimum, acknowledge the potential difference in sEPSC (freq), and explain the possible mechanisms of the difference in sEPSC frequency.

*Reviewer #2 (Recommendations for the authors):*

This study examines the effects of all-trans retinoic acid (atRA) on synaptic function and plasticity in dentate granule neurons of the murine hippocampus. Building on their previous work, the authors turn to systemic administration of atRA to examine its effects on synaptic function and plasticity in vivo. The data presented are very clear and the experiments appear to have been well executed. Systemic atRA had little effect on basal synaptic function of either excitatory or inhibitory inputs onto granule neurons, but significantly enhanced perforant path LTP induced by theta-burst stimulation and measured in vivo in anesthetized mice. These results are interesting and important, as they show that atRA can modulate plasticity in the absence of overt modulation of synaptic function. There are also a few limitations of this study, most of which are thoughtfully dealt with by the authors in the discussion.

---

## [Author Response]

Essential revisions:1) The authors claim that atRA does not have major effects on excitatory synaptic transmission in the DG neurons of hippocampus. However, the reviewers noticed that sEPSC frequency was increased significantly in the dorsal DG neurons. While we do not think more experiments are necessary, this change should not be ignored. Please provide some plausible explanations to this observation.

We thank the editors and reviewers for their positive evaluation of our study and for highlighting this important point. We went carefully through the manuscript and made several amendments to improve clarity by specifying “no major changes in excitatory synaptic strength are observed”. These findings are now substantiated with an additional dataset that we completed during the course of the revisions (see new figure 6). Assessment of synaptic cross sections in micrographs obtained from transmission electron microscopy showed no differences in basic morphological properties of pre- and postsynaptic compartments, i.e., PSD lengths and presynaptic vesicle counts, in vehicle-only controls and atRA-treated animals. Strikingly, a marked increase in the number of PSDs per area were observed in these experiments. These results show that atRA increases the density of asymmetric synapses in the molecular layer of the dentate gyrus in the dorsal hippocampus, thus providing an explanation for atRA mediated changes in sEPSC frequencies in that area.

“No major changes in synaptic transmission were observed in the ventral hippocampus while a significant increase in both sEPSC frequencies and synapse numbers were evident in the dorsal hippocampus 6 hours after atRA administration.”

“Next, transmission electron microscopy was used to assess the structural properties of excitatory synapses in the outer two thirds of the molecular layer in the dorsal hippocampus which is the layer of the major excitatory input from the entorhinal cortex (Figure 6). Cross sections of asymmetric synapses, i.e., the numbers and length of postsynaptic densities (PSD) and presynaptic vesicle counts, were quantified in control and atRA-treated mice (Figure 6A). It is well-established that PSD length in synaptic cross sections correlates to synaptic strength [34]. In agreement with our electrophysiological recordings, which showed no significant difference in the sEPSC amplitudes between the groups (c.f., Figure 1D), PSD lengths did not significantly change in the atRA-treated group (Figure 6B). However, a robust increase in the number of PSDs per area was detected, and presynaptic vesicle counts were not significantly different between the two groups (Figure 6B, C). These results indicate that the structural properties of synapses are not affected by atRA, and that increased synapse numbers may explain the increased sEPSC frequencies in the dorsal hippocampus of atRA-treated mice.”

“In the present study, however, we did not observe changes in excitatory synaptic strength in dentate granule cells in either the ventral or the dorsal hippocampus. Specifically, no changes in the sEPSC amplitudes were observed [12]. Consistent with these findings no major changes in the ultrastructural properties of excitatory synapses, i.e., PSD lengths and presynaptic vesicle counts of asymmetric synapses, were observed between the two groups. Interestingly, ultrastructural analysis revealed an increase in the number of asymmetric synapses in the dorsal hippocampus of atRA-treated animals.”

“Nevertheless, the results of the present study demonstrate increased sEPSC frequencies and synapse numbers in the dorsal hippocampus of atRA-treated animals, thereby confirming that atRA targets excitatory synapses in the dorsal hippocampus.”

2) Please specify the part of DG analyzed in in vivo study as dorsal and ventral DG neurons responded differently in slice physiology.

In vivo recordings were performed in the dorsal DG.

“Then a tungsten recording electrode (TM33B01KT, World Precision Instruments) was lowered in 0.1 mm increments while monitoring the waveform of the field excitatory postsynaptic potential (fEPSP) in response to 500 µA test pulses until the granule cell layer in the dorsal part of the hippocampus was reached (1.7-2.2 mm below the surface).”

“To test the effects of atRA on the ability of neurons to express synaptic plasticity, long-term potentiation (LTP) experiments on perforant path synapses to dentate granule cells were carried out in the dorsal hippocampus of anesthetized mice (Figure 7A).”

3) Compared to cortical neurons (from your previous publication) and many other studies on hippocampal pyramidal neurons, hippocampal DG neurons seem to be more resistant to the synapse-enhancing effects of atRA. While more experiments would help resolve whether granule cells are truly resistant to atRA's effects on basal synaptic function (or whether this is a route of administration/concentration issue), we think the primary conclusion is well supported by data presented. Thus, no additional experiments are required at this point, but the authors may want to make this point more clearly in the discussion.

We followed this suggestion and further discussed this point– also considering the findings of our EM analysis – and extended the following paragraph in the revised version of our manuscript stating more clearly that atRA indeed influences excitatory neurotransmission onto dentate granule cells of the dorsal hippocampus.

“These findings support the notion that atRA may assert its effect on synaptic transmission in a brain region- or cell type-specific manner. However, differences in the respective experimental settings must be carefully considered, such as the use of distinct tissue preparations (acute slices vs. organotypic tissue cultures vs. dissociated neurons) and differences in atRA administration (systemic vs. local; in vivo vs. ex vivo). Consequently, additional research is required to better understand the distinct effects of atRA on synaptic transmission and to determine, for example, possible concentration-dependent effects, the impact of a single dose vs. repeated (long-term) administration, and the cell-type specific responsiveness to atRA. Nevertheless, the results of the present study demonstrate increased sEPSC frequencies and synapse numbers in the dorsal hippocampus of atRA-treated animals, thereby confirming that atRA targets excitatory synapses in the dorsal hippocampus.”

We thank the editor and reviewers once again for their comments that helped us to improve the clarity and the quality of our study. We hope that this manuscript will now prove suitable for publication in *eLife*.

Reviewer #1 (Recommendations for the authors):The authors need to, at a minimum, acknowledge the potential difference in sEPSC (freq), and explain the possible mechanisms of the difference in sEPSC frequency.

We now acknowledge this finding in the abstract of the revised manuscript.

“No major changes in synaptic transmission were observed in the ventral hippocampus while a significant increase in both sEPSC frequencies and synapse numbers were evident in the dorsal hippocampus 6 hours after atRA administration.”

Reviewer #2 (Recommendations for the authors):This study examines the effects of all-trans retinoic acid (atRA) on synaptic function and plasticity in dentate granule neurons of the murine hippocampus. Building on their previous work, the authors turn to systemic administration of atRA to examine its effects on synaptic function and plasticity in vivo. The data presented are very clear and the experiments appear to have been well executed. Systemic atRA had little effect on basal synaptic function of either excitatory or inhibitory inputs onto granule neurons, but significantly enhanced perforant path LTP induced by theta-burst stimulation and measured in vivo in anesthetized mice. These results are interesting and important, as they show that atRA can modulate plasticity in the absence of overt modulation of synaptic function. There are also a few limitations of this study, most of which are thoughtfully dealt with by the authors in the discussion.

We thank the reviewer for this comment and have addressed this issue in the revised discussion of the manuscript as follows:

“In the present study, however, we did not observe changes in excitatory synaptic strength in dentate granule cells in either the ventral or the dorsal hippocampus. Specifically, no changes in the sEPSC amplitudes were observed [12]. Consistent with these findings no major changes in the ultrastructural properties of excitatory synapses, i.e., PSD lengths and presynaptic vesicle counts of asymmetric synapses, were observed between the two groups. Interestingly, ultrastructural analysis revealed an increase in the number of asymmetric synapses in the dorsal hippocampus of atRA-treated animals.”

“These findings support the notion that atRA may assert its effect on synaptic transmission in a brain region- or cell type-specific manner. However, differences in the respective experimental settings must be carefully considered, such as the use of distinct tissue preparations (acute slices vs. organotypic tissue cultures vs. dissociated neurons) and differences in atRA administration (systemic vs. local; in vivo vs. ex vivo). Consequently, additional research is required to better understand the distinct effects of atRA on synaptic transmission and to determine, for example, possible concentration-dependent effects, the impact of a single dose vs. repeated (long-term) administration, and the cell-type specific responsiveness to atRA. Nevertheless, the results of the present study demonstrate increased sEPSC frequencies and synapse numbers in the dorsal hippocampus of atRA-treated animals, thereby confirming that atRA targets excitatory synapses in the dorsal hippocampus.”